# Progerinin, an optimized progerin-lamin A binding inhibitor, ameliorates premature senescence phenotypes of Hutchinson-Gilford progeria syndrome

So-mi Kang[1,7], Min-Ho Yoon[1,7], Jinsook Ahn[2], Ji-Eun Kim[3], So Young Kim[4], Seock Yong Kang[4], Jeongmin Joo[4], Soyoung Park[1], Jung-Hyun Cho[1], Tae-Gyun Woo[1], Ah-Young Oh[1], Kyu Jin Chung[3], So Yon An[5], Tae Sung Hwang[5], Soo Yong Lee[6], Jeong-Su Kim[6], Nam-Chul Ha[2], Gyu-Yong Song[3] & Bum-Joon Park [1✉]

Previous work has revealed that progerin-lamin A binding inhibitor (JH4) can ameliorate pathological features of Hutchinson-Gilford progeria syndrome (HGPS) such as nuclear deformation, growth suppression in patient's cells, and very short life span in an in vivo mouse model. Despite its favorable effects, JH4 is rapidly eliminated in in vivo pharmacokinetic (PK) analysis. Thus, we improved its property through chemical modification and obtained an optimized drug candidate, Progerinin (SLC-D011). This chemical can extend the life span of $Lmna^{G609G/G609G}$ mouse for about 10 weeks and increase its body weight. Progerinin can also extend the life span of $Lmna^{G609G/+}$ mouse for about 14 weeks via oral administration, whereas treatment with lonafarnib (farnesyl-transferase inhibitor) can only extend the life span of $Lmna^{G609G/+}$ mouse for about two weeks. In addition, progerinin can induce histological and physiological improvement in $Lmna^{G609G/+}$ mouse. These results indicate that progerinin is a strong drug candidate for HGPS.

[1] Department of Molecular Biology, College of Natural Science, Pusan National University, Busan, Korea. [2] Department of Food Science, College of Agricultural Science, Seoul National University, Seoul, Korea. [3] Department of Pharmacy, College of Pharmacy, Chungnam National University, Daejeon, Korea. [4] New Drug Development Center, Daegu-Gyeongbuk Medical Innovation Foundation, Daegu, Korea. [5] Institute of Animal Medicine, College of Veterinary Medicine, Gyeongsang National University, Jinju, Korea. [6] Cardiovascular Center, Research Institute for Convergence of Biomedical Science and Technology, Pusan National University Yangsan Hospital, Yangsan, Korea. [7] These authors contributed equally: So-mi Kang, Min-Ho Yoon. ✉email: bjpark1219@pusan.ac.kr

Hutchinson-Gilford progeria syndrome (HGPS) is a very rare genetic disease[1–3]. It is also known as a premature aging syndrome[4,5]. A single point mutation in LMNA (G608G) generates progerin (50 amino acid deleted lamin A) by abnormal splicing[6–9]. Lamin A is matured through a four-step modifications. First, a 15-carbon farnesyl lipid is added to the carboxyl terminal cysteine by farnesyltransferase. Second, Zmpste24 protease cleaves the last three amino acids. Third, the newly exposed farnesylcysteine is carboxyl-methylated by a prenylprotein-specific methyltransferase. Finally, Zmpste24 removes terminal 15 amino acids of the protein, resulting in the release of mature lamin A[10–12]. Since Zmpste24-cleavage site is located in the deleted 50 AA region in progerin, progerin retains farnesylated C-terminus[12]. Based on this fact, lonafarnib, a farnesyltransferase inhibitor (FTI), has been suggested as a putative drug for HGPS[13]. Indeed, clinical trial of lonafarnib for HGPS has shown that it can extend the life span for ~2 years, although the trial has been performed without a placebo (single side experiment). However, in in vitro experiment, lonafarnib has cytotoxic effects, leading to the formation of donut-shaped nuclei[14] and apoptosis[15,16] rather than having a favorable effect. These results suggested more effective therapeutic approaches were needed for HGPS patients without severe side effect.

Here, we report that a binding inhibitor of lamin A (LA) and progerin called progerinin, improves aging-associated alterations in both in vitro and in vivo HGPS models, and consequently suggest that progerinin can be an effective treatment strategy for patients with HGPS.

## Results

### SLC-D011 ameliorates the premature aging characteristics of HGPS.
To test whether abnormal processing and retaining of farnesylation in progerin is critical for nuclear deformation in HGPS. We generated a point mutation to the CAAX motif of LA and progerin to mimic farnesylation inhibitory activity (Figs. S1a and S1b). We monitored nuclear morphologies of GFP-tagged LA, LA-C661A (LA-C661A), progerin (PG), and progerin-C611A (PG-C611A) positive HEK293 cells. However, farnesylation-deficient mutants did not show obvious difference with wild-type LA and progerin (Fig. 1a and Figs. S1c and S1d). The non-farnesylated progerin showed deformed nuclei to the similar extent as progerin. Wild-type LA and LA-C661A also represented similar percentage of nuclear deformation (Fig. S1e). Indeed, progerin and progerin-C611A reduced the expression of H3K9me3 (Fig. 1b and Fig. S1f and S1g), a marker of senescence[17]. We also observed increases in binding affinity of progerin, progerin-C611A, and LA-C661A (Fig. S1h). The biological meaning of such increase of LA-C661A is currently unclear. These results suggest that farnesylation in progerin is not critical for nuclear abnormality.

Our recent study has suggested that nuclear abnormality, a featured phenotype of HGPS cells, might be caused by strong binding between lamin A and progerin[17]. Indeed, an inhibitor (JH4) of progerin for lamin A binding can ameliorate nuclear deformation in HGPS cells and restore senescence-related markers such as p16/INK4A, DNA-PK, and H3K9me3 expression[17]. Moreover, treatment with JH4 via intraperitoneal injection (i.p) can extend the life span of LmnaG609G/G609G mouse for ~4 weeks (~25% increase)[17]. To use this chemical as a therapeutic drug, we tested its chemical property. Considering that patients with HGPS are children who need to take medication for their whole life[18,19], intravenous injection (i.v) is not a suitable delivery method. Thus, we first checked the possibility of oral administration. However, the half-life of JH4 was extremely short after in vivo oral administration (undetectable level; Figs. S2a–S2c) that we could not calculate its bioavailability (B.A). To overcome this

problem, we generated various JH4 modified chemicals (Fig. S3a) to improve its in vivo stability. We tested effects of JH4 derivatives on progerin-lamin A binding by GST-pull-down assay and progerin expression analysis (Figs. S3a and S3b). Among them, JH010 and SLC-D011 chemicals could block the interaction of progerin and lamin A (Fig. 1c and Figs. S3c and S3d; chemical structures of them are depicted in Fig. S3e). In addition, both JH010 and SLC-D011 could induce H3K9me3 expression and reduce progerin expression in HGPS cells (Figs. 1d and 1e and Figs. S3f and S3g). We also found that progerin levels were decreased after treatment with SLC-D011 in other kinds of HGPS cells without alteration of mRNA expression of progerin (Fig. 1f and Figs. S3h and S3i). As expression of progerin were reduced by SLC-D011 without a reduction of the mRNA levels, we speculated that progerin may lose its stability by dissociation from lamin A after treatment with SLC-D011. Indeed, we could confirm that the expression of progerin was reduced by SLC-D011 and restored by incubation with N-acetyl-leucinyl-norleucinal, a proteasome inhibitor (Fig. S3j). We also tested that SLC-D011 degraded progerin by autophagy activation. We observed that SLC-D011 did not change the expression of LC3 and beclin 1, autophagy-related genes (Fig. S3k). We also confirmed that the expression of beclin 1 could be reduced by 3-MA, an autophagy inhibitor, but not by SLC-D011 (Fig. S3l) SA-β-gal staining, a standard senescence assay[20], also showed that SLC-D011 could block cellular senescence (Fig. S3m). We then observed the expression of CENP1, IL-6, IL-8, and BRCA1 at mRNA levels were restored after treatment with SLC-D011 (Fig. S3n). Treatment with SLC-D011 could also rescue the expression of LAP2α in HGPS cells (Fig. S3o). In normal fibroblasts derived from healthy 9-year-old person, treatment with SLC-D011 did not change significantly (Figs. S3n and S3o). Moreover, JH010 and SLC-D011 could promote cell proliferation of HGPS cells (Fig. 1g and Fig. S4a) and ameliorate nuclear abnormality (Fig. 1h and Fig. S4b). Compared to normal fibroblasts, treatment with SLC-D011 increases cell proliferation in three different HGPS cells (Fig. S4c). We also observed a reduction of progerin level and an increase of H3K9me3 in three kinds of HGPS cells after treatment with JH4, JH010, or SLC-D011 (Figs. S4d and S4e).

### Therapeutic effect of SLC-D011.
In pharmacokinetic (PK) analysis, JH010 showed satisfactory bioavailability (B.A) result (~70%; Fig.2a and Fig. S5a). However, in in vitro absorption, distribution, metabolism, and excretion (ADME) analysis (Figs. S5b–S5e), JH010 inhibited hERG (human Ether-à-go-go-Related Gene), an ion channel (~75% inhibition at 1 μM; Fig. 2b). Thus, we performed in vitro ADME test for SLC-D011. SLC-D011 showed a B.A similar to JH010 (Fig. 2c and Fig. S6a) without severe hERG inhibition (Fig. 2b). Based on other indicators such as CYP inhibition, plasma stability, liver microsomal stability, and plasma protein binding, it appeared that SLC-D011 did not cause serious adverse effects (Figs. S6b–S6e). Thus, we focused on the therapeutic effect of SLD-D011. To check the efficacy of SLC-D011, we measured progerin expression in HGPS cells and found that 100 nM was enough for eliminating progerin (Fig. 2d, Figs. S7a–S7c). Exogenous progerin expression could be reduced by treatment with SLC-D011 (from 100 nM; Fig. 2e and Fig. S7d). Indeed, 100 nM of SLC-D011 seemed to be enough for inducing Ki67 (a cell proliferation marker[21]) expression in HGPS cells (Fig. 2f, Figs. S7e and S7f). A direct cell counting assay also revealed that 100 nM SLC-D011 could promote cell proliferation and ameliorate nuclear abnormality (Figs. S7g and S7h).

### SLC-D011 improves premature aging features of HGPS model mouse.
Next, the effect of SLC-D011 was tested in a mouse model

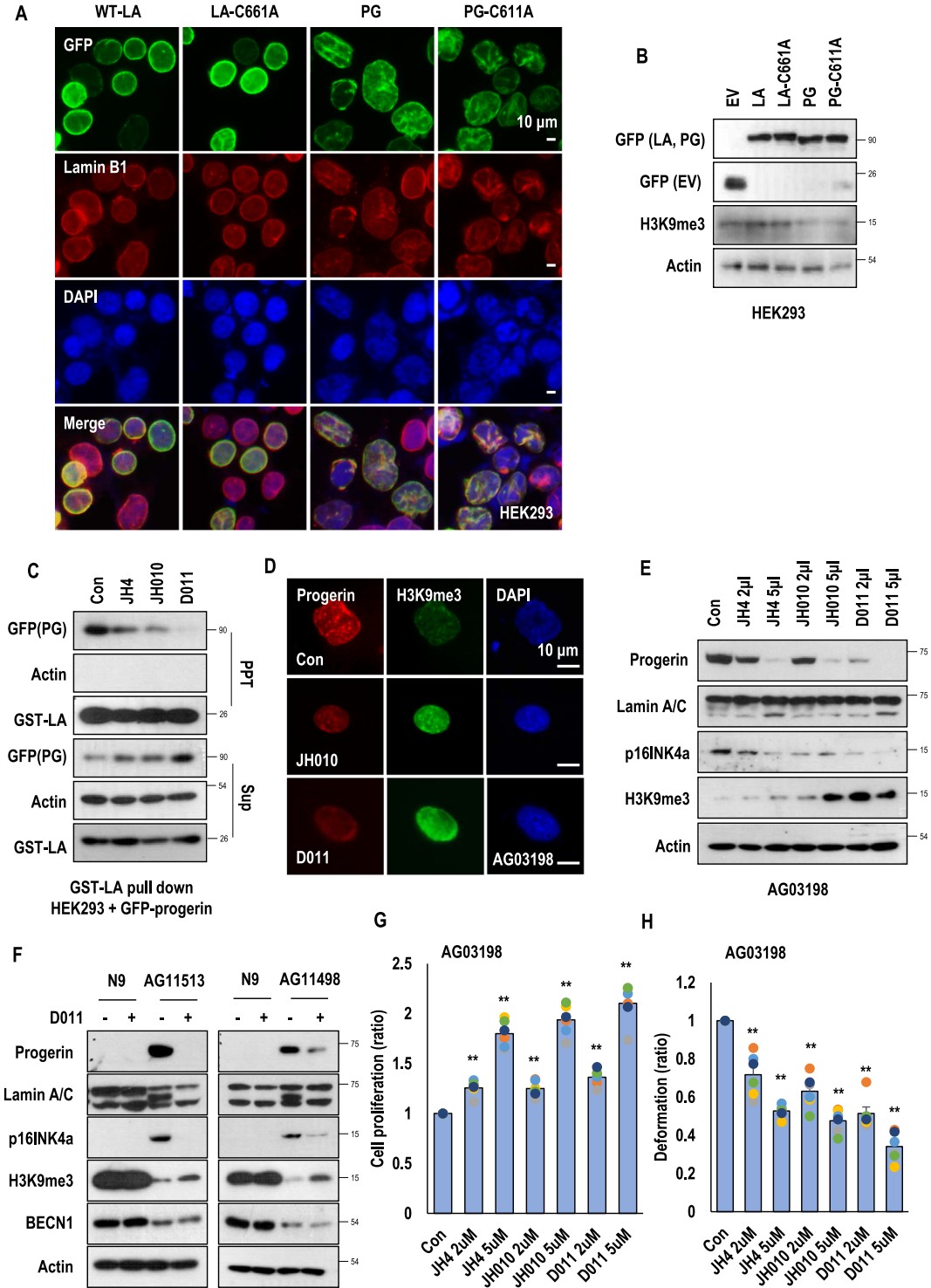

of progeria[22]. Treatment with SLC-D011 via i.p. injection (20 mg/kg, twice a week; Fig. 3a and Fig. S8) increased body weight (Fig. 3b) and extended life span of *Lmna*[G609G/G609G] mouse up to 21 weeks (Fig. 3c). Gross morphology was also obviously improved after treatment with SLC-D011 (increase of body size; Fig. 3a). A more interesting feature was obtained for *Lmna*[G609G/+] mice (Fig. 3d and Fig. S8). Treatment with SLC-D011 increased their body weights (Fig. 3e) and extended their life span from 46.7 weeks to 63 weeks (>1 year; Fig. 3f). This is a very exciting result because SLC-D011 can extend the life span of *Lmna*[G609G/+] for ~16 weeks longer than the control (untreated)

(Fig. 3f). We also observed improved morphologies such as the status of hair and body size of mouse (Fig. 3d). These results indicate that SLC-D011 is a strong drug candidate for HGPS. Thus, we chose SLC-D011 as a final drug candidate and renamed it as progerinin (progerin inhibitor).

**Oral administration of SLC-D011 prevents progeroid phenotypes in HGPS mouse.** Progerinin (SLC-D011) has very low solubility in water, which is one barrier for oral administration. Thus, we screened various solvents and found that a monoolein-based solution

**Fig. 1 SLC-D011 ameliorates premature aging features of HGPS. A**. HEK293 cells were transiently transfected with expression vector encoding wild-type lamin A (WT-LA), lamin A-C661A (LA-C661A), progerin, or progerin-C611A. LA-C661A or progerin-C611A with point mutation in CaaX motif was predicted to be lack of farnesylation, indistinguishable from that of authentic WT-LA or progerin. GFP-conjugated empty expression vector was used as a negative control. Cells were visualized at 24 h after transfection. **B** Transient transfection of HEK293 cells with expression vector encoding progerin or progerin-G611A leads to reduced expression of H3K9me3 ($n = 3$ independent experiments; two-tailed Student's $t$ test). **C**. JH4 derivatives (JH010 and SLC-D011(D011)) inhibit lamin A and progerin interaction. For binding inhibition assay, bead-conjugated lamin A (GST-LA) was incubated with HEK293 cells, transiently transfected with GFP-progerin (PG), and treated with indicated chemicals. Actin was used as a loading control. PPT, co-precipitated materials with recombinant proteins; *Sup* supernatant. 5 μM of each chemical was used for binding assay. **D** JH010 and SLC-D011 induce H3K9me3 expression. H3K9me3 reduction is a well-known marker for premature aged cells. Thus, H3K9me3 expression was evaluated by immunostaining. Both JH010 and SLC-D011 could induce H3K9me3 expression in HGPS cells (AG03198, $n = 3$ independent experiments; two-tailed Student's $t$ test). **E** SLC-D011 obviously suppresses progerin expression. For progerin expression analysis, each chemical was used to treat HGPS cells (AG03198, Coriell Cell Repositories) for 7 days. Compared to JH4 or JH010, SLC-D011 obviously suppressed progerin expression ($n = 3$ independent experiments; two-tailed Student's $t$ test). **F** SLC-D011 reduces the expression of progerin in other HGPS cells (AG11513 and AG11498, Coriell Cell Repositories). Treatment with SLC-D011 (2 μM) for 7 days reduced the expression of progerin and p16/INK4A but induced the expression of H3K9me3. However, normal fibroblast (N9; GM00038, foreskin fibroblast from a 9-year-old healthy child) did not respond to SLC-D011 ($n = 3$ independent experiments; two-tailed Student's $t$ test). **G** JH4, JH010, and SLC-D011 promote cell proliferation. Cell viability was determined by MTT assay after incubation with chemicals for 7 days ($n = 3$ independent experiments; two-tailed Student's $t$ test). **H** JH4, JH010, and SLC-D011 show similar activity for ameliorating nuclear deformation. After incubation with chemicals for 7 days, cells were stained with progerin and H3K9me3 antibodies ($n = 3$ independent experiment; two-tailed Student's $t$ test), **$p < 0.001$. Data are reported as mean ± SD. "Con" means Dimethyl sulfoxide (DMSO)-treated control. The data are normalized to DMSO-treated cells.

was useful for dissolving progerinin (Fig. S9a). Monoolein-based solution has been used to increase intestinal absorption without causing toxicity[23], suggesting that it would be suitable as a carrier for progerinin. Despite heating and sonicating to making a clear solution (Fig. S9a), progerinin was very stable in the solution (Fig. S9b). In PK analysis, progerinin was well recovered in the blood (Fig. S9c) in a dose-dependent manner (Figs. S9c and S9d). Oral administration of progerinin (50 mg/kg, daily administration) into $Lmna^{G609G/G609G}$ mice ameliorated gross morphology (Fig. 4a), increased body weight (Fig. 4b), and extended the life span of $Lmna^{G609G/G609G}$ mice for 10 weeks compared to untreated mice (Fig. 4c). Progerin levels in tissues of $Lmna^{G609G/+}$ mice after treatment with progerinin (oral administration at 50 mg/kg daily for 8 weeks) were also determined. Gross morphology (hair of mice) was improved after treatment with progerinin for 4 weeks (Fig. S10a). Such treatment also increased body weight (Fig. S10b) and life span (from 49 weeks to 65 weeks; Fig. S10c). Obvious reduction of progerin and the increase of H3K9me3 were observed in progerinin-treated $Lmna^{G609G/+}$ mice compared with untreated $Lmna^{G609G/+}$ mice (Fig. 4d and Figs. S10d and S10e). Histological analysis was then performed by hematoxylin and eosin (H&E) staining or Masson Trichrome staining for several tissues of $Lmna^{+/+}$ and $Lmna^{G609G/+}$ mice (Figs. S11a–S11f). In general, fibrosis occurs in tissues such as the heart, liver, and lungs in aged mice[24–26]. As the Masson Trichrome staining procedure stains the collagen-rich fibrotic regions in blue, it is suitable for assessing and visualizing the extent of fibrosis in tissues[27]. Therefore, we performed Masson Trichrome staining to check the fibrosis in the heart, liver, and lungs of mice. It was confirmed that fibrosis was increased especially in the heart and liver of untreated $Lmna^{G609G/+}$ mice compared with $Lmna^{+/+}$ mice, which was confirmed to be reduced by progerinin (Figs. S11a–S11c). We also observed a relatively loose connection between blood vessel walls and tissues in the liver (Fig. S11b) and lungs (Fig. S11c) of untreated $Lmna^{G609G/+}$ mice compared with $Lmna^{+/+}$ mice and progerin-treated $Lmna^{G609G/+}$ mice. $Lmna^{G609G}$ transgenic mice are known to show defects in skin tissues[28]. In the present study, epidermis of foot pad skin was thickened and dermal connective tissue was enriched in progerinin-treated $Lmna^{G609G/+}$ mice (Fig. S11d). We observed bone marrow reduction and loss of spinal cord cells in vertebrae of untreated $Lmna^{G609G/+}$ mice (Fig. S11e). Defects of bone marrow and spinal cord in vertebral column of $Lmna^{G609G/+}$ mice were also reduced after treatment with progerinin (Fig. S11e). Staining of femur (thigh bone) from untreated $Lmna^{G609G/+}$ mice with H&E revealed trabecular bone loss compared with age-matched $Lmna^{+/+}$ mice. We observed that

progerinin could restore trabecular and cortical bone thickening (Fig. S11f). Moreover, progerinin could improve grip strength (Fig. 4e) and heart beat rate (Fig. 4f) known to be reduced in a mouse model of HGPS[29]. Acquisition of kyphosis and dental abnormality are characteristics of $Lmna^{G609G}$ transgenic mice[22,28,30]. The mean kyphosis index (KI) of wild-type mice was >4. Incisors were normal at 41 weeks of age. The mean KI of untreated $Lmna^{G609G/+}$ mice was lower than that of $Lmna^{+/+}$ mice. Such low level of KI was recovered after treatment with progerinin (Figs. S11g and S11h). $Lmna^{G609G/+}$ mice had abundant adipose tissues in the abdomen after treatment with progerinin (Figs. S11g). Untreated $Lmna^{G609G/+}$ mice showed abnormalities of incisors (Fig. 4g). Their lower incisors grew toward the palate because of malocclusion[28]. Progerinin ameliorated these abnormalities of $Lmna^{G609G/+}$ mice (Fig. 4g). All of $Lmna^{G609G/+}$ mice treated with progerinin showed histological improvements compared with untreated $Lmna^{G609G/+}$ mice. Our results indicate that progerinin (monoolein-based SLC-D011 solution) could be used as a medication for patients with HGPS.

Next, we performed a comparative study for lonafarnib and progerinin. Differentially from lonafarnib, SLC-D011 at a lower concentration reduced the expression of transfected progerin in HEK293 cells (Fig. S12a). To confirm this finding, we measured the expression level of progerin in HGPS cells after chemical treatment. Although responses of different HGPS cells to lonafarnib were different (such as a reduction of progerin by lonafarnib in AG03199 and an induction of H3K9me3 in AG03198), SLC-D011 commonly reduced progerin level but induced H3K9me3 and cyclin B1, a cell cycle marker (Fig. 4h). In addition, long-term treatment with lonafarnib induced cell death in both normal fibroblasts and HGPS cells (Fig. S12b). In life span analysis of $Lmna^{G609G/+}$ mice, lonafarnib only extended 2 weeks longer, whereas SLC-D011 could extend ~14 weeks longer (Fig. S12c). Finally, we investigated the effect of combination treatment of SLC-D011 with lonafarnib. In long-term treatment experiment (2-week-treatment), we observed decreased level of progerin but increased expression levels of H3K9me3 and cyclin B1 in SLC-D011-treated HGPS cells regardless of the presence of lonafarnib (Figs. S12d and S12e). However, co-treatment with lonafarnib seemed to interrupt the induction of cyclin B1 by SLC-D011 (Fig. S12e). Indeed, lonafarnib induced donut-shaped nuclei (Fig. S12f), similar to a previous report[14]. Considering these results, SLC-D011 (progerinin) seems to be a more suitable candidate drug for HGPS than lonafarnib. We tested the toxicity of SLC-D011 in dogs and rats and found that it had no severe toxicity at

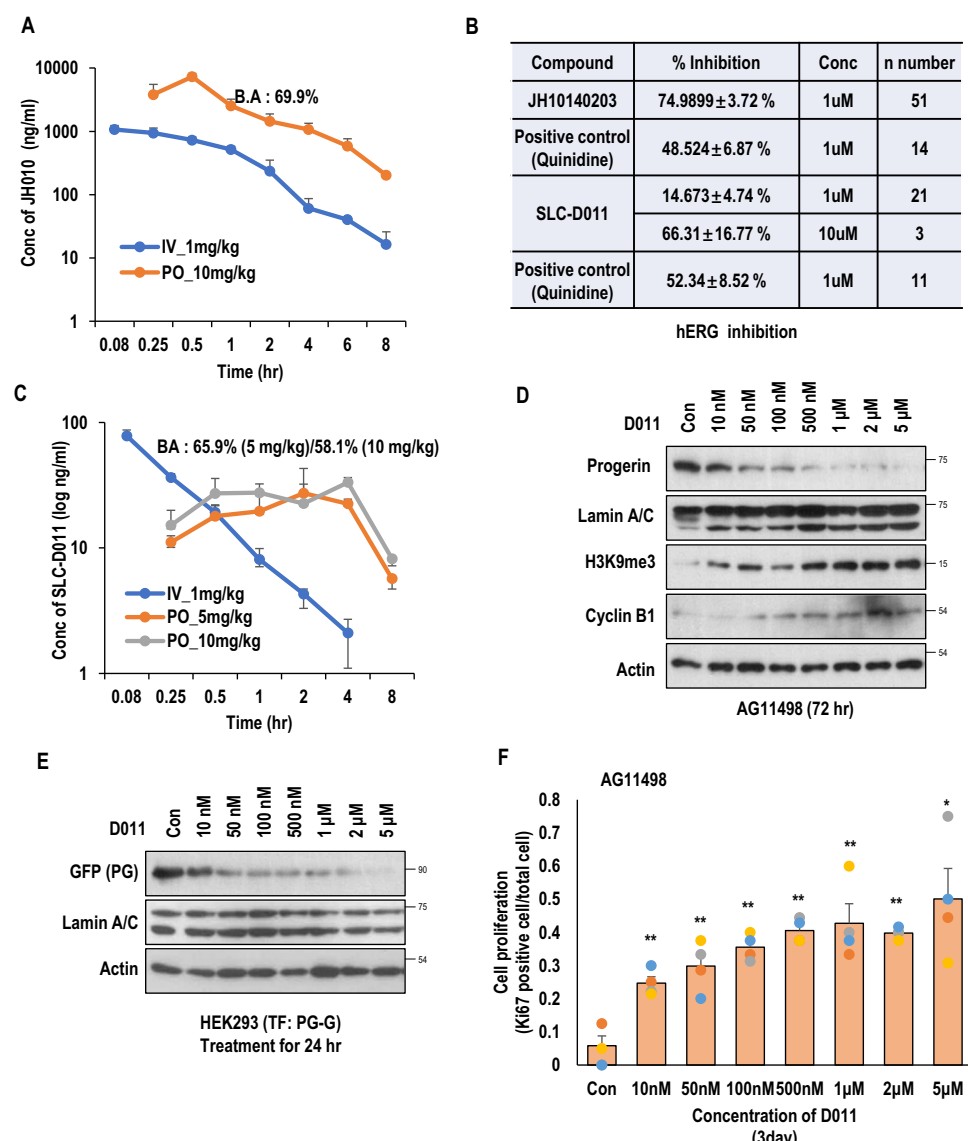

**Fig. 2 Therapeutic effect of SLC-D011. A**. In vivo pharmacokinetic (PK) analysis of JH010 (performed in rats). JH010 showed proper pharmacokinetic profile and very high Bioavailability (B.A). **B** Effect of SLC-D011 on human ERG (hERG). SLC-D011 did not show severe hERG inhibition. **C** In vivo pharmacokinetic analysis of SLD-D011. **D** Treatment with low concentration of SLC-D011 (100 nM) for 3 days reduces progerin expression but induces H3K9me3 and cyclin B1 expression in HGPS patient-derived cells (AG11498, Coriell Cell Repositories, $n = 3$ independent experiment; two-tailed Student's $t$ test). **E** SLC-D011 eliminates exogenous progerin expression. HEK293 cells transiently transfected with GFP-PG vectors were treated with SLC-D011 for 24 hr ($n = 3$ independent experiment; two-tailed Student's $t$ test). **F** SLC-D011 induces expression of Ki67 (a cell proliferating marker) in HGPS cells (AG11498, Coriell Cell Repositories). Ki67-positive cells were visualized at 3 days after treatment, and counted from photomicrographs ($n = 3$ independent experiment; two-tailed Student's $t$ test), **$p < 0.001$, *$p < 0.05$. "Con" means dimethyl sulfoxide (DMSO)-treated control.

very high concentrations (supplementary information1, page 10–11). Thus, we strongly suggest that progerinin (SLC-D011) is a very plausible drug for HGPS without adverse effects.

## Discussion

Understanding molecular mechanisms and pathological features of progerin is essential for developing clinical treatment for patients with HGPS. Over the past decade, active progress in HGPS research has led to an increasing number of treatment strategies[31–36]. Nevertheless, most of previous studies had insufficient preclinical data for transposition to patients with HGPS. In fact, farnesylation of progerin has been a main target for clinical drug development to prevent aging process in HGPS. Blocking farnesylation by (FTIs in HGPS cells can restore nuclear structure, cell proliferation, and chromatin organization[37,38]. In

addition, treatment with FTIs can enhance the growth and life span of *Zmpste24*-deficient mice[39]. Based on these studies, a clinical trial of an inhibitor of farnesyltransferase called lonafarnib has been performed for patients with HGPS[16,40,41]. Although it has some efficacies for HGPS, lonafarnib has several side effects. In addition, it does not work in all patients[42]. Lonafarnib also showed cytotoxic effects including the formation of donut-shaped nuclei and cell death in in vitro experiments after a long-term treatment (Figs. S12b and S1f; ref. [16–18]). Thus, we generated farnesylation-deficient (ASIM) progerin. It showed no difference compared with progerin (Fig. 1a and Figs. S1c). Why the ASIM progerin showed inconsistent results with previous reports[43,44] is unclear, but it is possible that eliminating protein farnesylation would not be sufficient to abolish the nuclear deformation, as some studies have suggested that

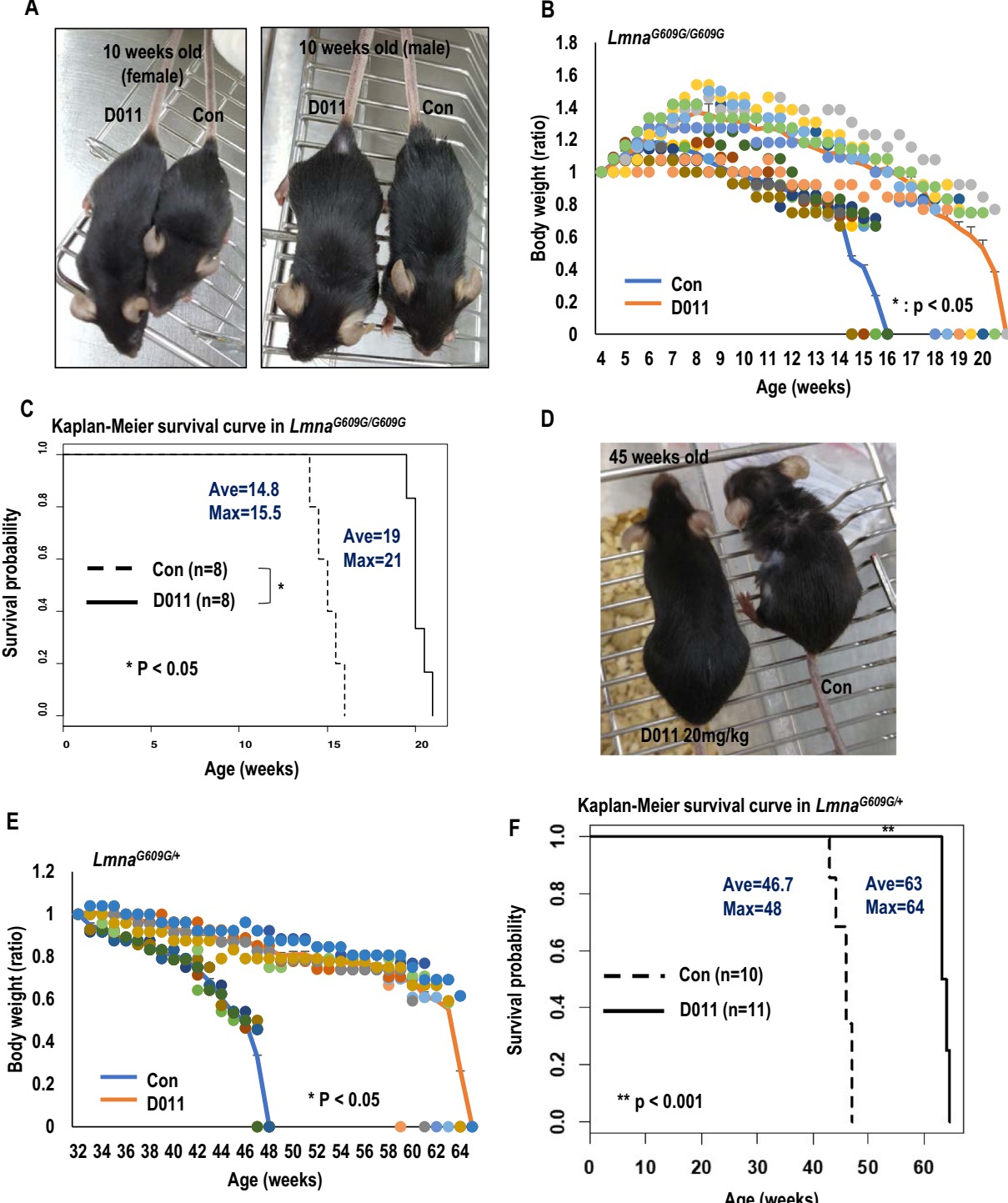

**Fig. 3 In vivo favorable effect of SLC-D011. A** Gross morphology of *Lmna^{G609G/G609G}* mice (10 weeks old) was ameliorated after injection with SLC-D011. At the same age, the experimental group (D011) of mice were larger than vehicle-injected control (Con) subjects regardless of gender. **B**, **C** SLC-D011 increases body weight **B** and extends life span **C** of *Lmna^{G609G/G609G}* progeria model mouse (Con: $n = 8$; SLC-D011: $n = 8$). Compared with 14.8 weeks of average (ave) life span with maximum (max) life span of 15.5 weeks, 20 mg/kg of intraperitoneal (i.p) injection (twice a week) of SLC-D011 could extend the life span upto 19 weeks (max = 21 weeks). *Lmna^{G609G/G609G}* mice were injected with SLC-D011 from age of 5-week old, *$p < 0.05$. **D** Gross morphology of SLC-D011 injected *Lmna^{G609G/+}* mouse. Sickly and weak features were observed in 45 weeks old *Lmna^{G609G/+}* mouse, but not observed in treated mouse, although they had the same age. **E** SLC-D011 increases body weights of *Lmna^{G609G/+}* mice (Con: $n = 10$; SLC-D011: $n = 11$), *$p < 0.05$. **F** Favorable effect of SLC-D011 on life span of *Lmna^{G609G/+}* mice. Compared with vehicle-injected control group (ave = 46.7 weeks and max = 48 weeks), SLC-D011 treatment obviously extended the average life span to 63 weeks (max = 64 weeks). The injection was started from 32 weeks old, **$p < 0.001$. "Con" means vehicle (a solvent composed of DMSO and PBS)-injected mouse group.

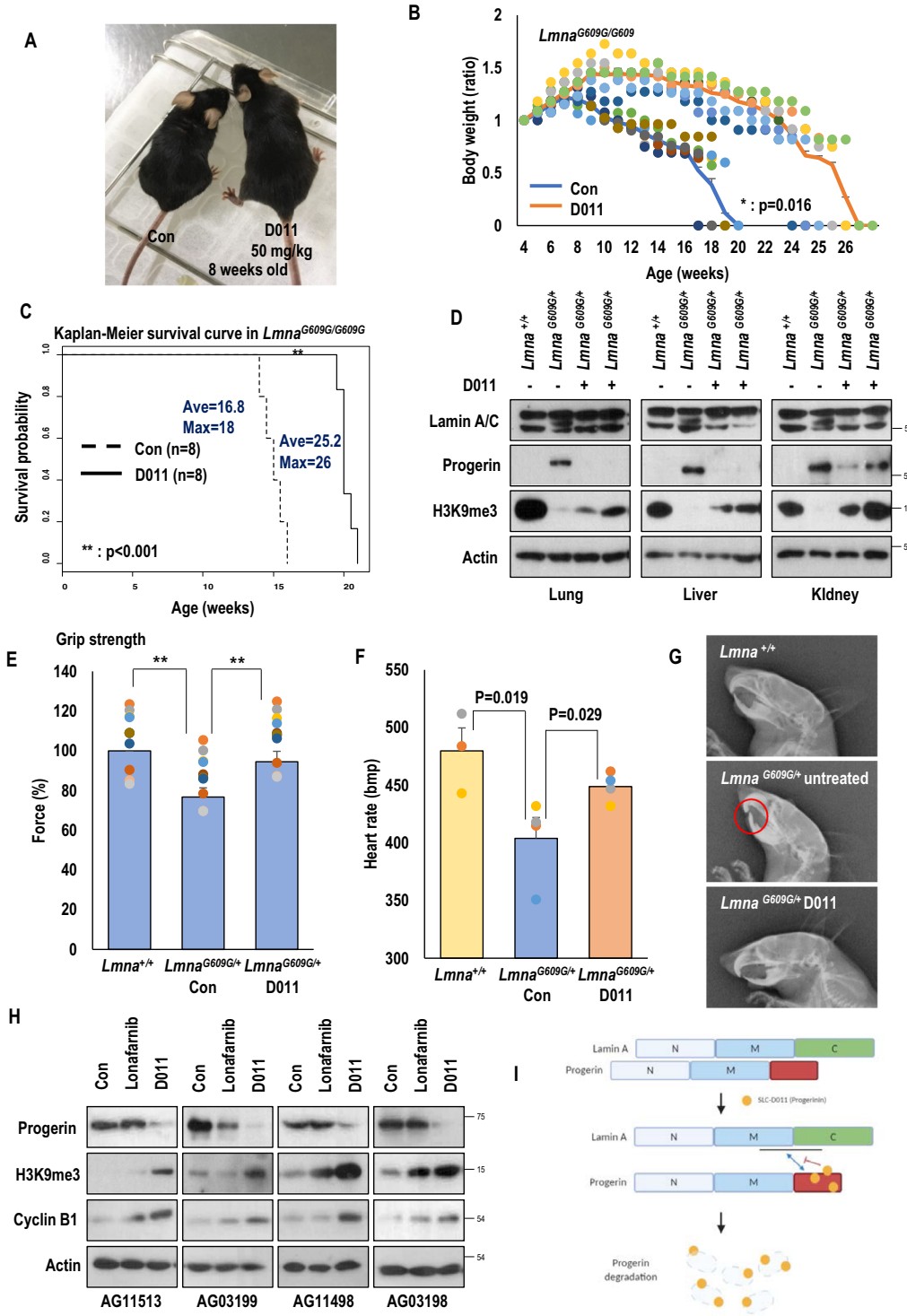

progerin could be geranylgeranylated when FTase activity is inhibited[45]. These results indicate that other therapies should be explored to cure patients with HGPS. Our previous study has shown that lamin A/C is the most important target of progerin through strong binding. Thus, we developed a novel drug called JH4 that could block the interaction between lamin A/C and progerin[17]. We confirmed a direct binding affinity of progerin for the middle region of lamin A and specific interaction of JH4 with progerin. JH4 alleviated nuclear defects and reversed senescence-related genes in HGPS cells. Moreover, administration of JH4 to $Lmna^{G609G}$ transgenic mice resulted in considerable improvement of progeroid phenotypes and life extension. However,

because of its rapid elimination in the body, JH4 was modified to a more stable form, SLC-D011 (also called progerinin). The C-terminal region of progerin was the direct target of progerinin (Fig. 4i) like JH4[17]. Interaction of progerinin with progerin interrupted the binding affinity of progerin to lamin A (Fig. 1c) and alleviated senescence features of HGPS cells including nuclear deformation and growth arrest (Figs. 1a–1h and Figs. 2d–2f). Moreover, administration of progerinin by i.p. injection ameliorated segmental progeroid phenotypes of $Lmna^{G609G}$ transgenic mice and extended the life span without toxicity (Fig. 3). However, it seemed that continuous injections caused stress to mice. Therefore, we administered progerinin through

**Fig. 4 Oral administration of SLC-D011 suppresses premature aging features. A** Gross morphology of SLC-D011-treated (D011) mice at 8 weeks old. Body size of treated mouse was apparently larger than that of the untreated control (Con) mouse. **B** Oral administration of SLC-D011 increases body weights of $Lmna^{G609G/G609G}$ mice. Compared with untreated control group, body weight of treated mouse was increased by ~35%, *$p < 0.05$. **C** Oral administration of SLC-D011 extends the life span of $Lmna^{G609G/G609G}$ mouse. SLC-D011 in monoolein-based solution was treated by oral gavage (50 mg/kg, daily) from 5 weeks old. Compared with untreated control group (ave = 16.8 weeks and max = 18 weeks), treated group showed extended life span (ave = 25.2 weeks and max = 26 weeks), **$p < 0.001$. **D** Reduction of progerin expression and increase of H3K9me3 expression in tissues of $Lmna^{G609G/+}$ mice after treatment with SLC-D011 by oral gavage (50 mg/kg, daily) for 8 weeks ($n = 3$ independent experiments; two-tailed Student's $t$ test). **E** Grip test of $Lmna^{G609G/+}$ mice at 41-week-old was performed after administration of SLC-D011 for 6 weeks. The test was repeated 10 times for each mouse ($n > 4$), **$p < 0.001$. Data are presented as mean ± SD. **F** Electrocardiographic analysis of $Lmna^{G609G/+}$ mice (untreated: $n = 4$; SLC-D011: $n = 4$) and $Lmna^{+/+}$ mice ($n = 3$) after treatment with SLC-D011 for 6 weeks (unpaired $t$ test). Heart rate is shown as beats per minute (bpm). **G** Dental abnormalities in $Lmna^{G609G/+}$ mice (untreated: $n = 4$; SLC-D011: $n = 5$) and $Lmna^{+/+}$ mice ($n = 4$). X-ray lateral projection of the skull of untreated LmnaG609G/+ mouse (untreated: $n = 4$; SLC-D011: $n = 5$) at 41 weeks of age compared to $Lmna^{+/+}$ mouse ($n = 4$) at same age shows abnormalities of incisors (unpaired $t$ test). Lower incisors of $Lmna^{G609G}$ transgenic mice grow toward the palate because of malocclusion. Oral administration of SLC-D011 for 6 weeks alleviates dental abnormalities. **H** Anti-aging effects of single treatment of SLC-D011 or lonafarnib were measured in HGPS patients-derived cells (AG11513, AG03199, AG11498, and AG03198, Coriell Cell Repositories). Single treatment of SLC-D011, compared to lonafarnib, reduced the expression of progerin but induced the expression of H3K9me3 and cyclin B1 in HGPS cells more effectively. HGPS patients-derived cells were treated with SLC-D011 (500 nM) or lonafarnib (500 nM) for 3 days. **I** Scheme of interaction between lamin A and progerin. SLC-D011 interrupts the interaction between lamin A and progerin through direct binding to the C-terminal region of progerin. "Con" and "untreated" mean vehicle (monoolein-based solution)-treated mouse group or vehicle (DMSO)-treated cells.

---

feeding. Progerinin was fed constantly and could be maintained in the mouse's body without stress. Oral administration of progerinin relieved the stress of mice and resulted in better anti-aging effects including life extension than i.p. injection (Figs. 4a–4c and Figs. S10a–S10c). Results of the present study suggest that progerinin is more stable and effective than other potential drugs. Thus, progerinin might be a promising treatment for children with HGPS without causing serious adverse effects.

## Methods

**Animal experiments.** Animal experiments were performed in a facility certified by the Association for Assessment and Accreditation of Laboratory Animal Care in compliance with animal policies approved by Pusan National University. The mouse work was performed under the study protocol PNU-2019-2181, as approved by the Institutional Animal Care and Use Committee. $Lmna^{G609G/609G}$ mice were generated by timed mating of heterozygous $Lmna^{G609G/+}$ provided by Carlos López-Otín (Universidad de Oviedo, Asturias, Oviedo, Spain). SLC-D011 was mixed with dimethyl sulfoxide (DMSO) and phosphate-buffered saline (PBS). It was then intraperitoneally injected into mice (20 mg/kg twice per week from 5-week-old). SLC-D011 and lonafarnib were orally administrated to mice daily at a concentration of 10 mg/ml in monoolein-based solution. Lonafarnib was generously provided by Merck, the Progeria Research Foundation (PRF) and the PRF Lonafarnib Pre-clinical Drug Supply Program. Control mice were treated with monoolein-based solution alone in the same way. $Lmna^{G609G/609G}$ mice were treated with clear chemical solution throughout the life span, starting from 5 weeks of age. $Lmna^{G609G/+}$ mice were treated via intraperitoneal and oral administrations, starting from 32 weeks of age.

**Cell culture and reagents.** Human fibroblast cells from HGPS patients (AG03198, 10-year-old female; AG03199, 10-year-old female; AG11513, 8-year-old female; AG11498, 14-year-old male), and normal person (GM00038, 9-year-old female N9) were obtained from Coriell Cell Repositories (Camden, NJ, USA) and maintained in Eagle's minimal essential medium supplemented with 15% fetal bovine serum, and 2 mM glutamine without antibiotics. HEK293 cells were obtained from the American Type Culture Collection (ATCC, Manassas, VA, USA) and maintained in liquid medium containing 10% FBS, and 1% penicillin–streptomycin at 37 °C with 5% $CO_2$.

**Chemical synthesis and characterization data.** Synthetic scheme of SLC-D011 is provided in the Supplementary information (2, page 1) file. The molecular structure of SLC-D011 was investigated using $^1$H-NMR Spectroscopy, $^{13}$C-NMR spectroscopy, mass spectrometry, and X-ray diffractometry (Supplementary information 2, page 1–4).

**Antibodies and reagents.** Antibodies used for experiments included anti-GFP (1:1000; sc-9996; Santa Cruz Biotechnology, Dallas, TX, USA); anti-GST (1:5000; sc-138; Santa Cruz Biotechnology), anti-His (1:1000; 66005-1-lg; Proteintech, Rosemont, IL, USA), anti-Actin (1:10000; sc-47778; Santa Cruz Biotechnology), anti-Lamin A/C (1:10000; sc-376248; Santa Cruz Biotechnology), anti-Progerin (1:100; sc-81611; Santa Cruz Biotechnology), anti-Progerin (1:300; ab66587; Abcam, Cambridge, UK), anti-Ki67 (1:200; Ab15580; Abcam), anti-cyclin B1

(1:100; sc-594; Santa Cruz Biotechnology), and anti-p16-INK4A (1:500; 10883-1-AP; Proteintech).

**Purification of recombinant proteins.** To obtain selenomethionyl-labeled protein for crystallization, *Escherichia coli* strain B834 (DE3; Novagen, USA) harboring plasmids encoding lamin A/C fragments (residue 250–400 or 556–664) were cultured in M9 medium supplemented with L-(+)-selenomethionine. Protein expression was induced by 0.5 mM isopropyl β-d-1-thiogalactopyranoside at 30 °C. Cells were harvested by centrifugation and resuspended in lysis buffer containing 20 mM Tris-HCl (pH 8.0) and 150 mM NaCl. Cells were disrupted with a sonicator. Cell debris was then removed by centrifugation. The supernatant of middle region fragment (residue 250–400) was loaded onto Ni-NTA affinity agarose resin (Qiagen, The Netherlands), pre-incubated with lysis buffer while the supernatant of C-terminal region fragment (residue 556–664) was loaded onto glutathione affinity agarose resin (Qiagen, The Netherlands), pre-incubated with lysis buffer. Target proteins were eluted with lysis buffer supplemented with 250 mM imidazole. These eluted fractions were further purified using anion-exchange chromatography (Hitrap Q HP, GE Healthcare, Chicago, IL, USA).

**Transfection of expression vectors.** GFP tag conjugated EV (GFP-EV), lamin A (GFP-LA), lamin A-C661A (GFP-LA-C661A), progerin (GFP-PG), and progerin-C611A (GFP-PG-C611A) vectors were used. GFP-PG and GFP-LA expression vectors were kindly provided by Misteli T. (National Cancer Institute [NCI], Frederick, MD, USA). GFP-LA-C661A and GFP-PG-C611A expression vectors were created by a single point mutation in CaaX motif. The CSIM sequence of lamin A, or progerin was mutated to ASIM. jetPEI (Polyplus Transfection, NY, USA) was used for transfection of expression vectors. Vectors were mixed with 1.5 μl of jetPEI in 150 mM NaCl solution. The mixture was added to HEK293 cells. After 4 h of incubation, medium was replaced with a new medium supplemented with 10% FBS.

**Immunoblotting.** Protein was extracted from cells using a radioimmunoprecipitation assay (RIPA) buffer (50 mM Tris-Cl, pH 7.5, 150 mM NaCl, 1% NP-40, 0.1% SDS, and 10% sodium deoxycholate). Samples were separated by sodium dodecyl sulfate-polyacrylamide gel electrophoresis (SDS-PAGE) and transferred to polyvinylidene difluoride (PVDF) membranes. Blotted membranes were blocked with 3% skimmed milk in TBS-T buffer (20 mM Tris pH 7.6, 150 mM NaCl, and 0.05% Tween 20) for 1 h followed by incubation with specific primary antibodies. Horseradish peroxidase (HRP)-conjugated goat anti-mouse, goat anti-rabbit, and mouse anti-goat IgG antibodies (Pierce, Thermo Fisher Scientific, Inc., Rockford, IL, USA) were used as secondary antibodies. Peroxidase activity was detected by chemiluminescence using an ECL kit (Intron, Seoul, Korea) following the manufacturer's instructions. Bands were quantified using Image J software (National Institute of Health, NIH).

**Western blot for tissue samples.** Lamin A/C and progerin amounts in tissue samples including lung, liver, and kidney analysis of two $Lmna^{+/+}$, four untreated $Lmna^{G609G/+}$, and four D011-treated $Lmna^{G609G/+}$ mice were evaluated by western blotting. Proteins were extracted with RIPA lysis buffer supplemented with protease inhibitors. Tissues were sonicated and centrifuged at 15,000 rpm for 15 min at 4 °C. Protein concentration was determined by Coomassie-blue staining. Cellular lysates in SDS-containing sample buffer were inactivated by heating at 98 °C for 7 min. Proteins were loaded onto 8% SDS-PAGE gels and transferred to

PVDF membranes. These membranes were blocked with 3% skimmed milk in TBS-T buffer for 1 h and incubated with primary antibodies including mouse monoclonal anti-progerin (1:100; sc-81611; Santa Cruz Biotechnology), mouse monoclonal anti-lamin A/C (1:500; MANLAC1; Developmental Studies Hybridoma Bank), and rabbit polyclonal anti-H3K9me3 (1;2000; Ab8898; Abcam) at 4℃ overnight. Blots were then incubated with 1:10,000 goat anti-mouse (Pierce, Thermo Fisher Scientific) or 1:10,000 goat anti-rabbit (Pierce, Thermo Fisher Scientific) IgG (HRP conjugated) in 1% skimmed milk and washed with TBS-T buffer. Bands were developed using and Intron ECL detection system and quantified using Image J (National Institute of Health, NIH).

**RNA isolation and RT-PCR.** For RT-PCR, total cellular RNA was extracted using RNA extraction kit (Qiagen). Gene expression studies were performed using cDNA synthesized from total RNA with MMLV RT (Invitrogen, Carlsbad, USA) and random hexamers. PCR from genomic DNA was perform using DiaStar Taq DNA polymerase (SolGent, Daejeon, Korea) and 50 ng of cDNA. Wild-type prelamin A cDNA was synthesized by PCR amplification using a forward primer, 5′-AAG-GAGATGAACCTGCTCCATC-3′ and a reverse primer, 5′-TTTCTTTGGCTTCAA GCCCCC-3′. The thermal cycling conditions were as follows: 94℃ for 3 min (activation), 94℃ for 1 min (denaturation), 61℃ for 1 min (annealing) and 72℃ for 42 sec (extension) up to 35 cycles. The PCR products were analyzed through 2% agarose electrophoresis and DNA was visualized by ethidium bromide staining and UV photography. Levels of other transcripts were also performed with the following oligonucleotide primers: CENP-A, 5′-ACAAGGTTGGCTAAAGGA-3′ and 5′-ATGCTTCTGCTGCCTCTT-3′; BRCA1, 5′-AGAGTGTCCCATCTGTCTGG-3′ and 5′-CGCTGCTTTGTCCTCAGAG-3′; IL-6, 5′-AAATGCCAGCCTGCTGAC-GAAC-3′ and 5′-AACAACAATCTGAGGTGCCCATGCTA-3′; IL-8, 5′-TGGCA GCCTTCCTGATTTCTG-3′ and 5′-AACTTCTCCACAACCCTCTGC-3′ and GAPDH 5′-ATCTTCCAGGAGCGAGATCCC-3′ and 5′-AGTGAGCTTCCCGT TCAGCTC-3′.

**Protein–protein interaction analyses.** For the analysis of protein–protein interaction, glutathione S-transferase (GST) pull-down assay and His pull-down assay were performed. To detect the interaction between wild-type lamin A and mutant lamin A, GST or His-bead-fused lamin A recombinant protein was incubated with GFP-tagged lamin A-C661A (GFP-LA-C661A), progerin (GFP-progerin), or progerin-C611A transfected HEK293 cells for 30 min at room temperature (RT). After washing once with PBS, precipitated materials were collected and subjected to SDS-PAGE and western blot analysis with anti-GFP and anti-GST.

**Immunofluorescence staining.** Cells were cultured on coverslips, washed with PBS, fixed with 4% paraformaldehyde (PFA) for 30 min at RT, and then permeabilized with 0.2% Triton X-100 at RT for 5 min. After treatment with blocking solution (anti-Human antibody diluted 1:400 in PBS) for 1 h, cells were incubated with anti-lamin A/C (1:400), anti-progerin (1:100), Ki67 (1:200), and anti-H3K9me3 (1:200) in blocking buffer overnight at 4 ℃. Finally, cells were incubated with fluorescein isothiocyanate and rhodamine-conjugated secondary antibodies at 4 ℃ for 7 h. Nuclei were stained with DAPI (4, 6-diamidino-2-phenylindole) at RT for 10 min. After cells were washed three times with PBS, coverslips were mounted with mounting solution (H-5501; Vector Laboratories, Burlingame, CA, USA). Immunofluorescence signal was detected with a fluorescence microscopy (Zeiss and Logos).

**MTT assay.** To determine the cellular viability, cells were treated with lonafarnib or SLC-D011 for 2 weeks. For MTT assay, cells were incubated with 0.5 mg/ml of 3-(4,5-dimethythiazol-2-yl)-2,5-diphenyl tetrazolium bromide (MTT) solution for 6 hr at 37 ℃. After removing MTT solution, the precipitated materials were dissolved in 200 μM DMSO and quantified by measuring the absorbance at 540 nm.

**Physiological analysis.** For analysis of heart rate, mice were anesthetized with 2.5% isoflurane and monitored using LOGIQ E9 (GE Healthcare). Data processing and analysis were performed using a TOMTEC system in Cardiovascular Center, Pusan National University Yangsan Hospital (Yangsan, Korea). For grip strength test, a tension meter measuring force was stationed horizontally on the platform. Mouse was allowed to grip the tension bar with front paws before being pulled slowly away from the bar until its grip was broken. Four female $Lmna^{G609G/+}$ mice, four $Lmna^{G609G/+}$ male mice, three female $Lmna^{+/+}$ mice, and three male $Lmna^{+/+}$ mice (all adults at 41 weeks old) were used for this test. Each mouse was subjected to the test ten times to determine its forelimb strength. Each estimate given corresponded to total reads measured ten times, excluding the minimum and the maximum from each group of animals.

**Radiological examinations.** The radiographic study was performed in anaesthetized living mice using a human radiographic system (College of Veterinary Medicine, Gyeongsang National University, Korea). A single radiograph of right lateral view of the whole body was obtained. Age-matched (at 41 weeks old) $Lmna^{+/+}$ mice, untreated $Lamn^{G609G/+}$ mice, and progerinin-treated $Lmna^{G609G/+}$ mice were radiographed. The images were recorded in DICOM format and then transferred to a personal computer.

KI[28,30] and abnormalities of the incisors were evaluated. Data processing and analysis were performed using a RadiAnt DICOM system in Institute of Animal Medicine, College of Veterinary Medicine, Gyeongsang National University (Jinju, Korea).

**Histology analysis.** Tissue specimens were fixed in 4% PFA and embedded in paraffin. Paraffin blocks were sectioned and transferred onto adhesive-coated slides. After deparaffinization and rehydration, tissue sections were stained with Masson Trichrome. Histology analysis was performed by BioLead Inc. (Seoul, Korea).

**Nuclear deformation counting.** For nuclear deformation cell counting, immunofluorescence staining was performed with lamin A or progerin antibodies. After staining, abnormal nuclear membrane was counted in randomly selected fields and expressed as percentages or actual numbers of total cells counted. Abnormalities of nuclear membrane were determined based on the following: (1) lamin A/C or progerin lining was extruded or engulfed, (2) having at least one bleb, and (3) irregular contour. Counting of cells with nuclear deformation was performed by three independent observers who were blinded to chemical treatment group. To analyze histone H3K9me3 intensity or expression, images were quantified using "color histogram" function of Image J software (National Institute of Health, NIH). Fluorescence intensities were subtracted by background signals.

**Chemical PK analysis and in vitro ADME test.** For PK analysis, 5 mg/kg of JH4 in 10% DMSO, 5% Tween 90, and 95% saline solution were intravenously injected and 10 mg/kg of JH4 in 10% NMP and 90% PEG400 solution were orally delivered. At pre-set time-points, blood concentration of JH4 was determined by LC-MS/MS analysis[22,23]. Other chemicals were also tested using similar protocol. In vitro ADME studies (plasma protein binding, CYP inhibition, microsomal stability, plasma stability, and hERG inhibition) were performed by New Drug Development Center (Daegu, Korea) using standard protocols[24,25].

**Toxicity studies in rats.** The protocol and procedures involving the care and use of animals in this study was reviewed and approved by IACUC of QuBEST BIO prior to conduct (Approval no.: QBSIACUC-A18122). During the study, the care and use of animals will be conducted in accordance with all applicable guidelines of Animal Welfare Act. Eleven specific pathogen-gree Sprague-Dawley rats (~6 weeks old) were obtained from SAMTAKO Ltd. (Osan, Korea) and nine rats were used for the study. After overnight fasting (~16 h, food but not water should be withheld overnight), test article formulations were dosed using plastic disposable feeding needle attached to a plastic disposable syringe. Food was withheld for a further 3–4 h after dosing. Each dose was based on the most recent body weight of each animal and the dose volume was 10 mg/kg. The dosing day was designated as Day 1. All animals were observed twice daily for mortality and moribundity during the study. A clinical observation was performed for all animals at the time of dosing and approximately 1, 2, and 4 h post-dose on dosing day, and once daily during 7-day observation period. Observations were included, but are not limited to, changes in the skin, fur, eyes and mucous membrane; respiratory, circulatory, autonomic and central nervous system function; somatomotor activity and behavior patterns. Individual body weights were measured for all animals on the day of animal receipt, randomization, prior to dosing start (Day 1) and study period (Days 4 and 7). Those animals survived on completion of the 7-day observation period were weighed body weight prior to necropsy and CO₂ inhalation anesthesia and exsanguinated from the abdominal aorta. In order to avoid autolytic change, a complete gross pathology examination of the carcass was performed as soon as possible after euthanasia of all animals. Necropsy was consisted of an external examination, including identification of all clinically recorded lesions, as well as a detailed internal examination. Toxicity studies were performed by QuBEST BIO Co., Ltd (Yongin, Korea).

**Toxicity studies in dogs.** The protocol and procedures involving the care and use of animals in this study was reviewed and approved by IACUC of QuBEST BIO prior to conduct (Approval no.: QBSIACUC-A18142). During the study, the care and use of animals will be conducted in accordance with all applicable guidelines of Animal Welfare Act. Non-naive one male and one female Beagle dogs (~16–24 months old, original supplier: ORIENTBIO Co., Ltd, Korea) were selected from the stock colony and assigned for the study. The route of administration was the oral (by gavage) route, which is the anticipated clinical route of exposure. Test article formulations were administered once via oral gavage. The day of 1st dosing was designated as Day 1. Each dose was administered via a syringe attached with 12-french feeding tube. A dose volume of 5 ml/kg was used and individual dose volumes were based on the most recent body weight. Each dose was followed by a distilled water of 5 ml. Each animal was observed twice daily (a.m. and p.m.) for mortality and moribundity; findings were recorded as they were observed. Cage side observations were made for each animal once daily; abnormal findings were recorded. Detailed observations were made for each animal once prior to treatment; abnormal findings (ranked/graded, if appropriate) or an indication the animal appears normal was recorded. Body weights were measured prior to each dosing. Prior to each dosing and ~24 h post each dose, blood samples for hematology examination were collected from the via cephalic vein into tubes with

K$_2$EDTA anticoagulant and following parameters were examined (ADVIA®2120, Germany). Prior to dosing and ~24 h post each dose, blood was collected as the same method and frequency with hematology analysis using no anticoagulant and then serum was separated by centrifugation and stored at freezer until analyze (AU 400, Olympus, Japan; RAPIDCHEM 744 Na$^+$/K$^+$/Cl$^+$ Analyzer, SIEMENS, Germany). In terminal procedures, all surviving animals were returned to the stock colony without necropsy. Toxicity studies were performed by QuBEST BIO Co., Ltd (Yongin, Korea).

**Statistics and reproducibility**. Data were analyzed with an unpaired or paired two-sample Student's $t$ test. $P$ value < 0.05 was considered statistically significant. Error bars indicate standard deviation (SD). Data for all figures were represented as mean values ± SD of at least three replicates. Details of statistical analyses and number of replicates ($n$) can be found in the figure legends.

**Reporting summary**. Further information on research design is available in the Nature Research Reporting Summary linked to this article.

## Data availability
Toxicity analysis of chemicals is available in Supplementary Data 1. Full blots are shown in Supplementary Information. All other data that support the findings of this study are available from the corresponding author upon reasonable request.

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

## Acknowledgements
This work was supported by National Research Foundation of Korea (NRF) grant funded by the Korea government (MSIT) (NRF-2020R1A4A1019322 to B.J.P.; NRF-2020R1F1A1075370 to B.J.P.; and NRF-2017R1A2B2005851 to G.Y.S.), and the Progeria Research Foundation (Grant #PRF 2019-75 to B.J.P.).

## Author contributions
S.K., M.H.Y., J.A., S.P., J.H.C., T.G.W., A.Y.O., S.Y.A., and S.Y.L. performed the experiments. J.E.K., K.J.C., G.Y.S., S.Y.K., S.Y.K., and J.J. synthesized and offered the chemicals. S.K., M.H.Y., and B.J.P. conceived the experimental designs. S.K. and B.J.P. wrote the manuscript. S.K., T.S.H., J.S.K., N.C.H., and B.J.P. analyzed the data.

## Competing interests

The authors declare no competing interests.
