## [Peer Review File · Communications Biology]

Reviewers' Comments:

Reviewer #1:

Remarks to the Author:

The manuscript by So-mi Kang et al. examines the effects of an optimized JH4 derivative, Progerinin (SLC-D011) in Hutchinson-Gilford Progeria Syndrome (HGPS). The authors found that Progerinin blocks progerin-lamin A binding, reduces progerin levels and improves HGPS phenotype in vitro. Using a progeria mouse model, Progerinin extends life span of LmnaG609G/G609G and LmnaG609G/+ mice with increased body weight. In addition, progerinin suppresses the muscle weakness including heart. The authors state that progerinin is a strong drug candidate for HGPS. The findings from this manuscript might be of importance for researchers and clinicians involved in the study of HGPS and aging. However, this manuscript reports premature observations regarding the decrease in progerin levels by progerinin as well as the rescue of premature aging phenotypes of HGPS in vitro. Some conclusions in the manuscript are not rigorously demonstrated by the experimental data shown. The quality of the manuscript needs major improvements.

Major points:

1. Line 50: The statement: "These results suggested that farnesylation in progerin is not critical for nuclear abnormality and aging" requires further investigation. In fact, eliciting HGPS disease depends on the carboxyl-terminal mutation used to eliminate protein farnesylation and not only the absence of farnesylation (Shao H. Yang et al. Human Molecular Genetics. 2011). In addition, nuclear shape correction alone has only little relevance in exploring progeria, and this correction should not be considered as an evaluation endpoint (Mohamed X. Ibrahim et al. Science.2013).
2. Line 66: The data on the rescue of HGPS disease phenotypes, in vitro, upon Progerinin treatment need more experiments. H3K9me rescue is not enough. Especially, the authors need to check the senescence in HGPS cells after Progerinin treatment using, for example, senescence associated β -galactosidase assay. What is the extent of the proliferation, DNA damage and inflammatory phenotype compared to age-matched WT control? I would also test the rescue of additional cellular phenotypes (e.g. LAP2 α , Lamin B1, HP1 α ...).
3. Line 67: The mechanisms underlying Progerinin effects on progerin levels needs to be addressed and discussed. Does this treatment act at the transcriptional or translational level? Macroautophagy may account for this, since several studies suggested that progerin is degraded by autophagy activation (Cao et al. Sci Transl Med. 2011, Gabriel et al. Aging Cell. 2015, Harhour et al. EMBO Molecular Medicine. 2017), it is not clear from the presented data that it does.

Other points:

1. Graphs with quantifications and statistical analysis of the Western Blot experiments are missing and should be included.
2. Discussion section is missing.
3. It would be important to better describe the progerinin-progerin interaction domain.
4. I would suggest to discuss the differences in results obtained with the two administration routes of progerinin (PO vs IV).
5. Better explain the legend of Figure 1. Fig1B: why there is no band in the GFP (LA, PG) line after transfection with the GFP vector (EV?) while this band is present in the 2nd line using the same anti-GFP antibody. Fig1C: what is the control "Con" in this experiment?
6. Fig.4D: The bands below progerin are sometimes absent/present in the same conditions and the same tissue (Lung, Liver). Do the authors have an explanation?
7. Page 5, line 114: the correct figure is S11 and not S10. Histological images are at too low magnification to observe fibrosis. It would be important to compare them to the same tissue from untreated mice.

Reviewer #2:

Remarks to the Author:

Open-label clinical trials in children with HGPS have tested drugs that block the farnesylation of progerin (e.g., lovastatin, lonafarnib). Unfortunately, the effects on disease progression have been modest. New therapeutic targets and approaches are required. In a previous report, the authors identified compounds that reduce the association of progerin with lamin A. Cell culture studies showed that these compounds (e.g., JH4) reduced in vitro phenotypes associated with progerin toxicity (e.g., abnormal nuclear shape, senescence), and extended survival in a mouse model of progeria (LmnaG609G). Importantly, they showed that JH4 did not extend lifespan or affect progeria disease phenotypes in mice lacking *Zmpste24*, which results in the accumulation of farnesyl-prelamin A. These latter studies demonstrated that the compounds were specific for progerin.

In the current manuscript, the main goal of the authors was to identify chemical derivatives that were effective when administered orally (in the previous study, JH4 was injected IP). They generated and screened a number of compounds using their lamin A-progerin pull-down assay, and identified several potential candidates. They performed several in vitro assays (proliferation, western blotting, microscopy) to test the efficacy of the chemicals against progerin and also performed pharmacokinetic studies to compare drug bioavailability and half-life. The most promising compound (SLC-DO11) was tested in LmnaG609G mice. The compound improved body weights and survival in mice when injected IP (20 mg/kg, 2x per week) or given by oral gavage (50 mg/kg, 1x per day). The authors conclude that these results provide strong evidence for testing in children with HGPS.

1. The authors examined the effects of progerin (progerin-GFP) and non-farnesyl-progerin (progerin-C611A-GFP) expression on nuclear morphology. Based on GFP fluorescence microscopy, they report that both proteins had similar effects on nuclear shape. This is inconsistent with previous studies showing that the inhibition of progerin farnesylation by genetic approaches (e.g., Mallampalli et al PNAS 102:14416) or with drugs (e.g., Varela I et al, Nat Med 14:767) significantly reduces abnormal nuclear shape in cells. The reason(s) for the opposite results are unclear. Since these are transient transfections, one possibility is that the proteins are expressed at different amounts. Western blot should be performed to verify similar amounts of GFP-progerin expression. Another possibility is that GFP distribution may not be the best measure of nuclear shape. The authors should assess nuclear shape using an antibody against a nuclear membrane protein (e.g., Lap2; see Mallampalli). Also, when they assess nuclear shape, they should avoid counting cells that are undergoing active cell division, which can alter the results. Images in figures 1A and S1C both show "abnormal nuclei" in cells that are dividing.

2. The authors show in multiple western blots that some compounds (e.g., DO11, JH010) reduce progerin protein levels. The lowering in some cells is almost 100%. This is an important finding. To verify this finding, the authors should perform lamin A/C western blots in figures 2D and 2E to establish that the effects are specific for progerin (lamins A and C do not decrease). Also, the authors should perform quantitative PCR studies to determine if lower progerin transcript levels can account for the reduced progerin protein levels.

3. Treatment of mice with progerinin (DO11) significantly improved survival in LmnaG609G mice. Western blots were performed (figure 4D) to determine if progerinin reduced progerin levels in tissues. However, the progerin western blot showed multiple bands and it is unclear which one is actually progerin. A positive control sample should be included in a new western blot (e.g., cell extract). The authors should also identify the antibody used to detect progerin (the anti-human progerin antibody or the lamin A/C antibody).

4. It is difficult to interpret the effects of progerinin treatment on tissue fibrosis (figure S11)

without a normal control sample (Lmna wild type). Images of sections from an age matched control mouse should be included.

5. The authors add as "data not shown" that toxicity studies were performed in rats and dogs. The data should either be shown and the animal study approvals added to the Methods, or the statement deleted.

Minor comments.

1. Line 35. The description of the processing of prelamin A to lamin A omits two steps; please include.

2. Line 42. Include a citation to support the authors' statement that lonafarnib exhibits cytotoxicity.

3. In many of the figures, statistics appear to have been applied to microscopy images and western blot experiments. For example (line 316), "Cells were visualized 24 hr after transfection (n = 3 independent experiments; two-tailed Student's t-test)." However, no quantitative data is provided. The authors should show the results in a bar graph in the Supplement or report the numbers in the text. A few other examples include lines 318, 327, 330, 348, 351, etc.

4. The lamin A/C staining in figure S7D is barely detectable. The quality of the images should be improved.

Reviewer #4:

Remarks to the Author:

This study by Bum-Joon Park's lab and colleagues is a follow up of a previous story in which the authors identified compounds that block the interaction of lamin A/C with the mutant lamin A protein "progerin". These compounds are able to alleviate many of the phenotypes of progeria (HGPS) cells, including nuclear deformation, heterochromatin defects, and growth arrest/senescence, as well as improve healthspan and lifespan of progeria mice (HGPS mouse model). In this manuscript, the authors modified the old compound to make it more stable in vivo. One of the new compounds generated (SLC-D011), named progerinin (progerin inhibitor), not only blocks the interaction of progerin and lamin A, but also reduces significantly progerin levels and ameliorates progeria phenotypes in cells in vitro. In addition, SLC-D011 has good bioavailability in vivo and does not cause adverse effects. Importantly, SLC-D011 is efficient ameliorating disease phenotypes in progeria mice when delivered via intraperitoneal injection and orally (dissolved in monoolein). Lastly, the authors show that the benefits of progerinin are much more robust than those of lonafarnib, which shows limited benefit.

The study is well-conducted and straightforward, mirroring the approach utilized previously. The main novelty is that the new compound, progerinin, can have a beneficial effect in progeria mice when delivered orally, which would be of enormous benefit for translating into the clinical setting for treating HGPS patients. For the most part, the experiments are well-controlled, and the conclusions from the results are appropriate.

Some weaknesses are the writing of the manuscript, which needs improvement in some parts, and lack of explanation of some results. For instance, the reduction of fibrosis in heart, blood vessels, liver and lung in progerinin-treated mice is not evident in the images, and not explained in the text. A more thorough characterization of tissues in the progeria mice (vascular, bone) will provide a stronger support for the claim that progerinin is a better treatment compared to current strategies. In addition, the manuscript is missing discussion of the overall findings, or putting the findings in the context of where the field is in terms of treatments for HGPS. Also, in the

discussion, the authors could explain the mechanism behind progerin loss upon progerinin treatment (defined in previous publication).

Reviewers' comments:

Reviewer #1 (Remarks to the Author):

The manuscript by So-mi Kang et al. examines the effects of an optimized JH4 derivative, Progerinin (SLC-D011) in Hutchinson-Gilford Progeria Syndrome (HGPS). The authors found that Progerinin blocks progerin-lamin A binding, reduces progerin levels and improves HGPS phenotype in vitro. Using a progeria mouse model, Progerinin extends life span of *Lmna*^{G609G/G609G} and *Lmna*^{G609G/+} mice with increased body weight. In addition, progerinin suppresses the muscle weakness including heart. The authors state that progerinin is a strong drug candidate for HGPS.

The findings from this manuscript might be of importance for researchers and clinicians involved in the study of HGPS and aging. However, this manuscript reports premature observations regarding the decrease in progerin levels by progerinin as well as the rescue of premature aging phenotypes of HGPS in vitro. Some conclusions in the manuscript are not rigorously demonstrated by the experimental data shown. The quality of the manuscript needs major improvements.

Thank you for taking your time to review our paper.

Major points:

1. Line 50: The statement: "These results suggested that farnesylation in progerin is not critical for nuclear abnormality and aging" requires further investigation. In fact, eliciting HGPS disease depends on the carboxyl-terminal mutation used to eliminate protein farnesylation and not only the absence of farnesylation (Shao H. Yang et al. Human Molecular Genetics. 2011). In addition, nuclear shape correction alone has only little relevance in exploring progeria, and this correction should not be considered as an evaluation endpoint (Mohamed X. Ibrahim et al. Science.2013).

- **As you suggested, many papers proposed the possibility of FTI on HGPS treatment. However, large portion of literatures use *Zmpste24*^{-/-} mouse model (Ibrahim et al., also used this mouse model).**
- **But, classical human HGPS is not related with loss of *Zmpste24*.**
- **Actually, FTI did not show obvious therapeutic effect on human clinical study and our mouse study, using *Lmna*^{G609G} model.**

- In Yang's report, they actually focused on protein prenylation. Protein prenylation involves geranylgeranylation as well as farnesylation. But our focus is only protein farnesylation.
- In fact, we did not know exactly why $Lmna^{csmHG}$ mouse model (Yang et al.) did not show progeria phenotype.
- One possibility is that deletion of the isoleucine of the CaaX motif (CSIM→CSM) can inhibit both farnesylation and geranylgeranylation of progerin.
- We also guess, the CSM progerin may interrupt the interaction of lamin A binding. But it is not our focus.
- Our report only focuses on the absence of farnesylation. We want to show that abnormal processing and retaining of farnesylation in progerin may not be critical for nuclear deformation in HGPS cells. Indeed, there was no obvious difference between nuclear morphologies of progerin positive cells and farnesylation-deficient progerin (progerin-C611A) positive cells (Fig. 1A and Fig. S1C).
- We also observed that expression of H3K9me3, a marker of senescence, was reduced in progeria-C611A positive cells like as progerin positive cells (Fig. 1B).
- What we emphasize in this sentence is that eliminating farnesylation is not enough to reduce nuclear deformation. We do not think that nuclear correction is the end point of the evaluation. We think that nuclear morphology is one of representative part of various progeria explorations.
- In Ibrahim et al., they showed that increase of AKT signaling may suppress progeria, which would be related with AKT activation.
- Actually, AKT activity is mainly regulated by lipid related myristoylation (Katharina M. Siess et al. Biochem Soc Trans. 2019, M.H. Grider et al. J Neurosci Res. 2009)
- To make sure what we want to say, we changed the statement from "These results suggested that farnesylation in progerin is not critical for nuclear abnormality and aging" to "These results suggested that farnesylation in progerin is not critical for nuclear abnormality"

2. Line 66: The data on the rescue of HGPS disease phenotypes, in vitro, upon Progerinin treatment need more experiments. H3K9me rescue is not enough. Especially, the authors need to check the senescence in HGPS cells after Progerinin treatment using, for example, senescence associated β -galactosidase assay.

- To address your suggestion, we measured the SA-b-gal in 3 kinds of HGPS cells and observed that progerinin could suppress the SA-b-gal expression. Thus, we added this new result in Supplementary Figure 3M.

What is the extent of the proliferation, DNA damage and inflammatory phenotype compared to age-matched WT control? I would also test the rescue of additional cellular phenotypes (e.g. LAP2 α , Lamin B1, HP1 α ...).

- We confirmed that expression of several genes such as CENP1, IL6, IL8 and BRCA1 was altered at mRNA levels in HGPS cells compare to wild type (WT, normal fibroblasts) control (Fig. S3N). Expression levels of CENP1 and BRCA1 were reduced but IL6 and IL8 were increased in HGPS cells. Treatment with progerinin restored the levels of these gene similar to normal fibroblasts (Fig. S3N).
- The expression of LAP2 α is reduced in HGPS cells compared to WT control. We also observed the rescue of LAP2 α expression after treatment with progerinin (Fig. S3O).
- The extent of proliferation in HGPS cells were increased after treatment with progerinin. However, there was no obvious difference in normal fibroblasts after treatment with progerinin (Fig. S4C).

3. Line 67: The mechanisms underlying Progerinin effects on progerin levels needs to be addressed and discussed.

- The mechanisms of progerinin effects on progerin was addressed in the discussion section by referencing our previous report (SJ Lee et al, J Clin Invest, 2016) and adding new simple diagram (Fig. 4I).

Does this treatment act at the transcriptional or translational level?

- Since progerinin is working as protein-protein interaction inhibitor (SJ Lee et al, J Clin Invest, 2016), it should not alter the transcription level. Indeed, our RT-PCR result showed that progerin does not alter the progein expression (Fig. S3I)
- Instead, proteasome inhibitor (ALLN), but not autophagy inhibitors (3-MA), blocked the progerinin-induced progerin reduction, indicating that progerinin block the interaction of progerin with lamin A and promote progerin turn-over (Fig. S3J-L).
- Considering our result, progerinin regulates post-translational level. We described it in this version.

Macroautophagy may account for this, since several studies suggested that progerin is degraded by autophagy activation (Cao et al. Sci Transl Med. 2011, Gabriel et al. Aging Cell. 2015, Harhoury et al. EMBO Molecular Medicine. 2017), it is not clear from the presented data that it does.

- **We tested whether effects of progerinin were associated with autophagy activation. The expression of LC3 and beclin1 (BECN1), autophagy-related genes, were not changed after treatment with progerinin in a dose-dependent manner (Fig. S3K). In addition, it was confirmed that only 3-MA, an autophagy inhibitor, could lower the expression of beclin1 (Fig. S3L).**
- **Otherwise, Progerinin-induced progerin reduction is related with proteasomal degradation. In fact, proteasome inhibitor (ALLN) can block the reduction of progerin (Fig. S3J).**
- **Again, we suggested that progerinin promote degradation of progerin via proteasome.**

Other points:

1. Graphs with quantifications and statistical analysis of the Western Blot experiments are missing and should be included.

- **We added graphs with quantifications and statistical analysis of WB experiments in the section of supplementary figures (for example, Fig. S3F-H, Fig. S7C-E).**

2. Discussion section is missing.

- **According to your suggestion, we separated our text and added discussion section.**
- **Also, we enriched discussion part**

3. It would be important to better describe the progerinin-progerin interaction domain.

- **The progerinin-progerin interaction domain was addressed in the discussion section by attaching our previous report (SJ Lee et al, J Clin Invest, 2016) and a figure (Fig. 4I).**

4. I would suggest to discuss the differences in results obtained with the two administration routes of progerinin (PO vs IV).

- **It seemed that continuous injection caused stress to mice. However, progerinin could be fed constantly and maintained in a mouse's body without stress**

through oral administration. For this reason, PO creates better anti-aging effects than injection. We addressed it in discussion section.

5. Better explain the legend of Figure 1. Fig1B: why there is no band in the GFP (LA, PG) line after transfection with the GFP vector (EV?) while this band is present in the 2nd line using the same anti-GFP antibody.

→ The size gap of the GFP (LA, PG) and GFP-EV was large, so we displayed them separately. Raw data was attached to show that they came from the same membrane (Raw data page 1).

Fig1C: what is the control "Con" in this experiment?

→ "Con" means Dimethyl sulfoxide (DMSO)-treated control. We used DMSO as a solvent of chemicals in *in vitro* experiments. We also included this information at section of figure legends.

6. Fig.4D: The bands below progerin are sometimes absent/present in the same conditions and the same tissue (Lung, Liver). Do the authors have an explanation?

→ In previous experiment, tissue seemed to be not clearly dissolved, so that non-specific protein (maybe antibody) was reacted with secondary Ab.
**→ To address this, we used the sonicator and pre-clearing step by protein-A/G-
agrose bead.**
→ In this version, we obtained the better results without non-specific interaction (Fig. 4D)

7. Page 5, line 114: the correct figure is S11 and not S10. Histological images are at too low magnification to observe fibrosis. It would be important to compare them to the same tissue from untreated mice.

→ We corrected the part you mentioned. We also changed to higher magnification images including tissues from wild-type models (Fig. S11).

Reviewer #2 (Remarks to the Author):

Open-label clinical trials in children with HGPS have tested drugs that block the farnesylation

of progerin (e.g., lovastatin, lonafarnib). Unfortunately, the effects on disease progression have been modest. New therapeutic targets and approaches are required. In a previous report, the authors identified compounds that reduce the association of progerin with lamin A. Cell culture studies showed that these compounds (e.g., JH4) reduced in vitro phenotypes associated with progerin toxicity (e.g., abnormal nuclear shape, senescence), and extended survival in a mouse model of progeria (LmnaG609G). Importantly, they showed that JH4 did not extend lifespan or affect progeria disease phenotypes in mice lacking Zmpste24, which results in the accumulation of farnesyl-prelamin A. These latter studies demonstrated that the compounds were specific for progerin.

In the current manuscript, the main goal of the authors was to identify chemical derivatives that were effective when administered orally (in the previous study, JH4 was injected IP). They generated and screened a number of compounds using their lamin A-progerin pull-down assay, and identified several potential candidates. They performed several in vitro assays (proliferation, western blotting, microscopy) to test the efficacy of the chemicals against progerin and also performed pharmacokinetic studies to compare drug bioavailability and half-life. The most promising compound (SLC-DO11) was tested in LmnaG609G mice. The compound improved body weights and survival in mice when injected IP (20 mg/kg, 2x per week) or given by oral gavage (50 mg/kg, 1x per day). The authors conclude that these results provide strong evidence for testing in children with HGPS.

Thank you for taking your time to review our paper.

1. The authors examined the effects of progerin (progerin-GFP) and non-farnesyl-progerin (progerin-C611A-GFP) expression on nuclear morphology. Based on GFP fluorescence microscopy, they report that both proteins had similar effects on nuclear shape. This is inconsistent with previous studies showing that the inhibition of progerin farnesylation by genetic approaches (e.g., Mallampalli et al PNAS 102:14416) or with drugs (e.g., Varela I et al, Nat Med 14:767) significantly reduces abnormal nuclear shape in cells. The reason(s) for the opposite results are unclear.

- ➔ **According to *in vivo* studies (Shao H. Yang et al. J Clin Invest. 2008 and Shao H. Yang et al. Hum Mol Genet. 2011), the inhibition of progerin farnesylation by genetic approaches still shows disease phenotypes of progeria.**
- ➔ **These results are not matched with the results of Mallampalli's even they had same genetic modification (progerin's carboxyl-terminal –CSIM motif was changed to –SSIM).**

- Actually, we did not know clear reason of different results from ours. But, in my opinion, excessive expression of GFP-lamin A or progerin in Mallampalli's experiment seems to be one of reason.
- Indeed, we can easily detect the nuclear spotting or aggregation of lamin A or progerin, similar to Fig. 4A in Mallampalli's paper, when excessive transfection occurred.
- Concerning this, we are now preparing the separate paper.
- Analysis of drugs in a previous study (Varela I et al, Nat Med 14:767) was performed using *Zmpste24*-deficient mouse models. However, *Zmpste24*-deficient mouse is not fairly related to classical HGPS. Human progeria-like *Lmna*^{G609G} transgenic mouse is suitable for HGPS model.
- Therefore, we treated with FTI (lonafarnib) in *Lmna*^{G609G} mice and observed that FTI did not produce promising results (Fig. S12C).
- We also observed that treatment with FTI induced the formation of donut-shape nuclei and cell death in both normal fibroblasts and HGPS cells without ameliorating effect on nuclear abnormality (Fig. S12B and S12F). Based on these results, we think long-term administration will not be good for patients with HGPS.

Since these are transient transfections, one possibility is that the proteins are expressed at different amounts. Western blot should be performed to verify similar amounts of GFP-progerin expression.

- To address this critics, we performed cell counting and western blot for verifying the GFP-tagged protein expression and added a graph with quantification (Fig. S1D).

Another possibility is that GFP distribution may not be the best measure of nuclear shape. The authors should assess nuclear shape using an antibody against a nuclear membrane protein (e.g., Lap2; see Mallampalli). Also, when they assess nuclear shape, they should avoid counting cells that are undergoing active cell division, which can alter the results. Images in figures 1A and S1C both show "abnormal nuclei" in cells that are dividing.

- To ascertain it, we stained the lamin B1 and found that lamin B1 also showed the abnormality in progerin-transfected cells with or without farnesylation-defective mutation. In addition, cells seemed to be not dividing (Fig. 1A and Fig. S1C).

2. The authors show in multiple western blots that some compounds (e.g., DO11, JH010) reduce progerin protein levels. The lowering in some cells is almost 100%. This is an important finding. To verify this finding, the authors should perform lamin A/C western blots in figures 2D and 2E to establish that the effects are specific for progerin (lamins A and C do not decrease).

- **It is very nice suggestion. To address your critic, we re-analyzed the samples and added lamin A/C WB data.**
- **As you expected, lamin A/C expression was not affected by progerinin treatment.**

Also, the authors should perform quantitative PCR studies to determine if lower progerin transcript levels can account for the reduced progerin protein levels.

- **Since progerinin is working as protein-protein interaction inhibitor (SJ Lee et al, J Clin Invest, 2016), it should not alter the transcription level. Indeed, our RT-PCR result showed that progerin does not alter the progein expression (Fig. S3I)**

3. Treatment of mice with progerinin (DO11) significantly improved survival in *Lmna*^{G609G} mice. Western blots were performed (figure 4D) to determine if progerinin reduced progerin levels in tissues. However, the progerin western blot showed multiple bands and it is unclear which one is actually progerin. A positive control sample should be included in a new western blot (e.g., cell extract). The authors should also identify the antibody used to detect progerin (the anti-human progerin antibody or the lamin A/C antibody).

- **In previous experiment, tissue samples seemed to be not completely lysed, so that non-specific protein (maybe antibody) was reacted with secondary Ab.**
- **To address this, we used the sonicator and pre-clearing step by protein-A/G- agrose bead.**
- **In this version, we obtained the better results without non-specific interaction (Fig. 4D)**
- **As you also commented, we indentified the progerin WB data is actual progerin. We used anti-lamin A/C antibody (Manlac, Developmental Studies Hybridoma) to detect lamin A/C in tissues from mice. Using this antibody we could detect both of expression of lamin A/C and progerin in *Lmna*^{G609G} mice. We also detect the progerin WB using anti-progerin antibody (Santa Cruz 13A4D4). By those results, we confirmed progerinin could reduce the expression of progerin level in *Lmna*^{G609G} mice (Fig. 4D).**

4. It is difficult to interpret the effects of progerinin treatment on tissue fibrosis (figure S11) without a normal control sample (*Lmna* wild type). Images of sections from an age matched control mouse should be included.

- **We included age-matched control group samples (Fig. S11). We could observe that the tissue status of progerinin-treated *Lmna*^{G609G} mice improved similarly to that age-matched control mice.**
- **We also included results from experiments on other tissues such as foot pad skin, vertebra, femur, and incisors of mice (Fig. 4G and Figs. S11D-S11H).**
- **We could observe that progerinin improved conditions of skin layers of foot pad. It also increased bone density and cell proliferation in vertebra and femur of *Lmna*^{G609G} mice.**
- **By performing X-ray lateral projection, kyphosis (Fig. S11G) and abnormalities of incisors (Fig. 4G) were restored in progerinin-treated mice.**
- **Considering our result, progerinin could block the premature aging features in progeria mouse model.**

5. The authors add as “data not shown” that toxicity studies were performed in rats and dogs. The data should either be shown and the animal study approvals added to the Methods, or the statement deleted.

- **We included results of toxicity studies in supplementary information (attached original reports). We also added information of the animal study approvals and experimental procedures of toxicity studies to the Methods section.**

Minor comments.

1. Line 35. The description of the processing of prelamin A to lamin A omits two steps; please include.

- **As you mentioned, we included the missing parts.**

2. Line 42. Include a citation to support the authors’ statement that lonafarnib exhibits cytotoxicity.

- **To support our statement that lonafarnib exhibits cytotoxicity, we included citations and added our experimental results (Figs. S12B and S12F).**
- **We could observe treatment with lonafarnib in HGPS cells induced donut-shaped nuclei (Fig. S12F) and cell death (Fig. S12B).**

3. In many of the figures, statistics appear to have been applied to microscopy images and western blot experiments. For example (line 316), "Cells were visualized 24 hr after transfection (n = 3 independent experiments; two-tailed Student's t-test)." However, no quantitative data is provided. The authors should show the results in a bar graph in the Supplement or report the numbers in the text. A few other examples include lines 318, 327, 330, 348, 351, etc.

→ **We added graphs with quantifications and statistical analysis of WB experiments in the section of supplementary figures (for example, Fig. S3F-H, Fig. S7C-E).**

4. The lamin A/C staining in figure S7D is barely detectable. The quality of the images should be improved.

→ **In this figure, we focused on showing overall expression of Ki-67 level after treatment with progerinin. That's why we placed the images at low magnification.**

→ **As you'd commented, we also attached the lamin A/C stained figures at higher magnification (Fig. S7F). We could observe nuclear deformation was ameliorated after treatment with progerinin.**

Reviewer #3 (Remarks to the Author):

This study by Bum-Joon Park's lab and colleagues is a follow up of a previous story in which the authors identified compounds that block the interaction of lamin A/C with the mutant lamin A protein "progerin". These compounds are able to alleviate many of the phenotypes of progeria (HGPS) cells, including nuclear deformation, heterochromatin defects, and growth arrest/senescence, as well as improve healthspan and lifespan of progeria mice (HGPS mouse model). In this manuscript, the authors modified the old compound to make it more stable in vivo. One of the new compounds generated (SLC-D011), named progerinin (progerin inhibitor), not only blocks the interaction of progerin and lamin A, but also reduces significantly progerin levels and ameliorates progeria phenotypes in cells in vitro. In addition, SLC-D011 has good bioavailability in vivo and does not cause adverse effects. Importantly, SLC-D011 is efficient ameliorating disease phenotypes in progeria mice when delivered via intraperitoneal injection and orally (dissolved in monoolein). Lastly, the authors show that the benefits of progerinin are much more robust than those of lonafarnib, which shows limited benefit.

The study is well-conducted and straightforward, mirroring the approach utilized previously. The main novelty is that the new compound, progerinin, can have a beneficial effect in progeria mice when delivered orally, which would be of enormous benefit for translating into the clinical setting for treating HGPS patients. For the most part, the experiments are well-controlled, and the conclusions from the results are appropriate.

Thank you for taking your time to review our paper.

Some weaknesses are the writing of the manuscript, which needs improvement in some parts, and lack of explanation of some results. For instance, the reduction of fibrosis in heart, blood vessels, liver and lung in progerinin-treated mice is not evident in the images, and not explained in the text.

- **As you suggested, we added some improved explanations including citations. We exchanged some figures at higher magnification and improved explanations.**
- **The blue area of tissues indicates the collagen-rich fibrotic region. We observed that this region was reduced in progerinin-treated mice compare to untreated mice.**

A more thorough characterization of tissues in the progeria mice (vascular, bone) will provide a stronger support for the claim that progerinin is a better treatment compared to current strategies.

- **It is very nice suggestion. To address your comments, we also included results from experiments on other tissues such as foot pad skin, vertebra, femur, and incisors of mice (Fig. 4G and Figs. S11D-S11H).**
- **We could observe that progerinin improved conditions of skin layers of foot pad. It also increased bone density and cell proliferation in vertebra and femur of *Lmna*^{G609G} mice (Figs. S11D-S11H).**
- **By performing X-ray lateral projection, kyphosis and abnormalities of incisors were restored in progerinin-treated mice (Fig. 4G and Fig. S11G).**

In addition, the manuscript is missing discussion of the overall findings, or putting the findings in the context of where the field is in terms of treatments for HGPS.

- **According to your suggestion, we separated our text and added discussion section.**
- **Also, we enriched discussion part**

Also, in the discussion, the authors could explain the mechanism behind progerin loss upon progerinin treatment (defined in previous publication).

→ The mechanisms of progerinin effects on progerin was addressed in the discussion section by attaching our previous report (SJ Lee et al, J Clin Invest, 2016) and a figure (Fig. 4I).

Reviewers' Comments:

Reviewer #1:

Remarks to the Author:

The authors have answered my concerns.

Reviewer #2:

Remarks to the Author:

1. The authors examine the contribution of progerin farnesylation to nuclear shape deformation. Based on the microscopy studies (Figs. 1a, S1c, S1d), the authors conclude that farnesylation is not critical for nuclear abnormality. Considering that there are published studies suggesting the opposite conclusion (some groups using the same approach, such as Francis Collins PNAS 102:12879 and Susan Michaelis PNAS 102:14416), the authors need to provide the quantitative data supporting their conclusion. At a minimum, they should measure nuclear shape deformation in WT, lamin A, progerin, and progerin-C661A expressing cells in at least 100 GFP positive cells. Images of a few cells is not sufficient evidence. If the data is consistent with their conclusion, they should add a sentence or two in the Discussion that the reason(s) for the inconsistent results are unclear. Related to this, the authors are incorrect that the Varela study (Nat Med 14:767) only performed studies in Zmpste24 KO; they also examined nuclear shape in human HGPS cells.

2. The authors include new microscopy images (Figs. 1a and S1c). They are an improvement; however, the transfection efficiencies of the GFP-proteins in the new images, appear to be much less than the reported 70% in Fig. S1d. Are the images showing representative fields?

3. Additional information must be provided for the RT-PCR studies reported in Fig. S3i. The identity of the cells (presumably WT and HGPS human fibroblasts); the identity of the two bands in the gel (presumably prelamin A and progerin); the DNA markers. If transcript levels will be measured using ethidium bromide-stained gels (instead of the delta CT method), they need to report the number of cycles performed and document that the amplification is in the linear phase for each set of primers.

4. The authors included age-matched wild-type control samples in their new histology studies (Fig. 4g and Figs. S11d-S11h). Were any quantitative measurements done to support their conclusions that progerin improves tissue fibrosis, bone density, adiposity, bone marrow, trabecular bone, etc? If not, the numbers of animals examined should be stated (e.g., 3 of 5 animals examined showed an improvement).

Minor comments.

1. In Figs. 1g and 1h, it appears that the data are normalized to DMSO-treated cells. If this is correct, this should be stated in the Methods section or the legend to the figure.

2. The authors included an improved western blot by performing immune precipitation with protein AG beads (Fig. 4d). This modification should be included in the Methods. The profile of the lamin protein bands looks correct; however, I think the molecular weight marker is incorrectly placed. Progerin is not 54 kD.

3. In figures (many) where they do not show the quantitative data, they should delete the mention of statistical measurements "(n = 3 independent experiments; two-tailed Student's t-test)."

Reviewer #4:

Remarks to the Author:

The authors have extensively modified the manuscript, adding many new experiments to address the critiques from the three reviewers. They have also added a Discussion section where they put their data in the context of what is known in the field. The findings are highly significant for the field of progeria, and add a new potential therapeutic strategy.

Reviewers' comments:

Reviewer #1 (Remarks to the Author):

The authors have answered my concerns.

Thank you for your positive answer.

Reviewer #2 (Remarks to the Author):

1. The authors examine the contribution of progerin farnesylation to nuclear shape deformation. Based on the microscopy studies (Figs. 1a, S1c, S1d), the authors conclude that farnesylation is not critical for nuclear abnormality. Considering that there are published studies suggesting the opposite conclusion (some groups using the same approach, such as Francis Collins PNAS 102:12879 and Susan Michaelis PNAS 102:14416), the authors need to provide the quantitative data supporting their conclusion. At a minimum, they should measure nuclear shape deformation in WT, lamin A, progerin, and progerin-C661A expressing cells in at least 100 GFP positive cells. Images of a few cells is not sufficient evidence.

→ We measured the percentage of nuclear deformation counting at least 100 GFP positive cells and included a bar graph (Fig. S1E).

If the data is consistent with their conclusion, they should add a sentence or two in the Discussion that the reason(s) for the inconsistent results are unclear.

→ As you commented, we added a sentence in the Discussion section.

Related to this, the authors are incorrect that the Varela study (Nat Med 14:767) only performed studies in Zmpste24 KO; they also examined nuclear shape in human HGPS cells.

→ We know that well. In Varela study (Nat Med 14:767), they actually focused on protein prenylation including both farnesylation and geranylgeranylation. But our focus is only protein farnesylation. When we treated FTI in human HGPS cells, we observed that FTI induced the formation of donut-shape nuclei and cell death

without ameliorating effect on nuclear abnormality (Fig. S12B and S12F). Based on these results, we think long-term inhibition of FTase activity will not be good for HGPS patients.

2. The authors include new microscopy images (Figs. 1a and S1c). They are an improvement; however, the transfection efficiencies of the GFP-proteins in the new images, appear to be much less than the reported 70% in Fig. S1d. Are the images showing representative fields?

→ To ascertain it, we changed new images (Fig. 1A and Fig. S1C) showing representative fields.

3. Additional information must be provided for the RT-PCR studies reported in Fig. S3i. The identity of the cells (presumably WT and HGPS human fibroblasts); the identity of the two bands in the gel (presumably prelamin A and progerin); the DNA markers. If transcript levels will be measured using ethidium bromide-stained gels (instead of the delta CT method), they need to report the number of cycles performed and document that the amplification is in the linear phase for each set of primers.

→ We added the DNA markers (Fig. S3I) and reported PCR cycling conditions in Method section.

4. The authors included age-matched wild-type control samples in their new histology studies (Fig. 4g and Figs. S11d-S11h). Were any quantitative measurements done to support their conclusions that progerin improves tissue fibrosis, bone density, adiposity, bone marrow, trabecular bone, etc? If not, the numbers of animals examined should be stated (e.g., 3 of 5 animals examined showed an improvement).

→ We did not quantify the measurements of improvement in new histological studies. All of examined progerin-treated *Lmna*^{G609G/+} mice were improved. We stated it in the Result section.

Minor comments.

1. In Figs. 1g and 1h, it appears that the data are normalized to DMSO-treated cells. If this is correct, this should be stated in the Methods section or the legend to the figure.

→ **We stated in the legend to the figure (at the end of the Fig 1 legend).**

2. The authors included an improved western blot by performing immune precipitation with protein AG beads (Fig. 4d). This modification should be included in the Methods. The profile of the lamin protein bands looks correct; however, I think the molecular weight marker is incorrectly placed. Progerin is not 54 kD.

→ **Sorry for our mistake. As you mentioned, the molecular weight marker was incorrectly placed. We replaced it (Fig. 4D). Progerin was detected between 62kDa (lamin C) and 69kDa (lamin A) size in progeroid mouse cells.**

3. In figures (many) where they do not show the quantitative data, they should delete the mention of statistical measurements "(n = 3 independent experiments; two-tailed Student's t-test)."

→ **We edited the mention of statistical measurements. Measurements are specified only where quantitative data are indicated.**

Reviewer #4 (Remarks to the Author):

The authors have extensively modified the manuscript, adding many new experiments to address the critiques from the three reviewers. They have also added a Discussion section where they put their data in the context of what is known in the field. The findings are highly significant for the field of progeria, and add a new potential therapeutic strategy.

Thank you for your positive answer.

Reviewers' Comments:

Reviewer #2:

Remarks to the Author:

The authors have addressed my concerns.